Dex-Benchmark: datasets and code to evaluate algorithms for transcriptomics data analysis

http://orcid.org/0000-0002-8256-5878 Xie Zhuorui
Chen Clara
Ma’ayan Avi avi.maayan@mssm.edu
Pharmacological Sciences, Icahn School of Medicine at Mount Sinai , New York, NY , USA
Gomez Shawn
Electronic publication date: 2023 Nov 8
Publication date: 2023
Volume: 11
Electronic Location ID: e16351
Received 2023 Jun 2; Accepted 2023 Oct 4
Copyright: © 2023 Xie et al.
Copyright year: 2023
Copyright holder: Xie et al.
License: This is an open access article distributed under the terms of the Creative Commons Attribution License, which permits unrestricted use, distribution, reproduction and adaptation in any medium and for any purpose provided that it is properly attributed. For attribution, the original author(s), title, publication source (PeerJ) and either DOI or URL of the article must be cited.
License URL: https://creativecommons.org/licenses/by/4.0/

Keywords: Benchmarking, Differential expression, RNA-seq, Transcriptomics, Workflows, Signatures, Drug disocvery, Target discovery, Systems biology, Dexamethasone

Funding: National Institutes of Health (NIH) OT2OD030160, U24CA224260, U24CA264250, U24CA271114, R01DK131525, and RC2DK131995 This work was funded by the National Institutes of Health (NIH) grants numbers OT2OD030160, U24CA224260, U24CA264250, U24CA271114, R01DK131525, and RC2DK131995. The funders had no role in study design, data collection and analysis, decision to publish, or preparation of the manuscript.

==============================
Many tools and algorithms are available for analyzing transcriptomics data. These include algorithms for performing sequence alignment, data normalization and imputation, clustering, identifying differentially expressed genes, and performing gene set enrichment analysis. To make the best choice about which tools to use, objective benchmarks can be developed to compare the quality of different algorithms to extract biological knowledge maximally and accurately from these data. The Dexamethasone Benchmark (Dex-Benchmark) resource aims to fill this need by providing the community with datasets and code templates for benchmarking different gene expression analysis tools and algorithms. The resource provides access to a collection of curated RNA-seq, L1000, and ChIP-seq data from dexamethasone treatment as well as genetic perturbations of its known targets. In addition, the website provides Jupyter Notebooks that use these pre-processed curated datasets to demonstrate how to benchmark the different steps in gene expression analysis. By comparing two independent data sources and data types with some expected concordance, we can assess which tools and algorithms best recover such associations. To demonstrate the usefulness of the resource for discovering novel drug targets, we applied it to optimize data processing strategies for the chemical perturbations and CRISPR single gene knockouts from the L1000 transcriptomics data from the Library of Integrated Network Cellular Signatures (LINCS) program, with a focus on understudied proteins from the Illuminating the Druggable Genome (IDG) program. Overall, the Dex-Benchmark resource can be utilized to assess the quality of transcriptomics and other related bioinformatics data analysis workflows. The resource is available from: https://maayanlab.github.io/dex-benchmark.

Introduction

Over the past two decades, there has been a rapid rise in the availability of genome-wide gene expression data. Since 2003, the number of samples within the Gene Expression Omnibus (GEO) (Barrett et al., 2013) generated via transcriptomics techniques, including microarray and RNA-sequencing, has increased to over one million (Lachmann et al., 2018). At the same time, thousands of tools, algorithms, and workflows have been developed for analyzing such data. While some methods have become more popular, and new versions of some tools have been released, the community of users often applies the most popular rather than the most effective algorithms and tools to analyze and process their data. To mitigate this trend for maximally extracting knowledge from these data, more unbiased community benchmarks need to be developed. Objectively comparing the ability of tools to extract knowledge accurately and effectively from data can crucially assist in prioritizing tools and algorithms and lead to improved decision making in analysis pipelines (Pearson, 2008; Schurch et al., 2016; Su et al., 2014).

Past studies have provided results from benchmarking methods for specific tasks such as large-scale data integration (Luecken et al., 2022), RNA-seq data quantification (Everaert et al., 2017; Teng et al., 2016), and single cell RNA-seq analysis pipelines (Nguyen et al., 2023; Tian et al., 2019). For example, identifying and extracting differentially expressed genes from sequencing data is often a critical step for further downstream analyses, and there has been much focus on benchmarking various methods for computing differential gene expression (DGE) (Baik, Yoon & Nam, 2020; Quinn, Crowley & Richardson, 2018). Some of these studies evaluate methods by reproducing existing results (Seyednasrollah, Laiho & Elo, 2015), while others analyze simulated data (Robles et al., 2012; Soneson & Delorenzi, 2013) or use quantitative reverse transcription PCR (qRT-PCR) data as a gold standard for more systematic benchmarking (Rapaport et al., 2013; Zhang et al., 2014). Prior work has also focused on specific research contexts, such as the analysis of inter-species RNA-seq datasets (Bastide et al., 2023), or the application of various existing quantification and differential expression approaches to proteomics data (Lin et al., 2022). While these previous efforts have helped to generate a foundational set of guidelines for transcriptomics analysis pipelines, most of these benchmarking efforts are implemented in R, and are limited in scope to commandline scripts available at the time of the original publication. Hence, the benchmarking code itself often requires a high level of coding experience to run, reducing the generalizability of the benchmarking methods for researchers who are interested in evaluating the same methods on different types of data. However, there are some efforts to make such analyses more accessible. For instance, the R package compcodeR (Soneson, 2014) enables users to generate an HTML report comparing and visualizing results from different DGE analysis methods but is limited to synthetic count data. Another resource, RNAontheBENCH (Germain et al., 2016), provides benchmarks for genome alignment and quantification in addition to DGE analysis. However, while the R source code is still available, the web interface for this application is no longer supported.

Moreover, there have been few attempts at systematically benchmarking analysis methods for alternative gene expression profiling datasets such as the L1000 data, which was produced as part of the NIH Library of Integrated Network-Based Cellular Signatures (LINCS) program (Keenan et al., 2018; Xie et al., 2022). The L1000 assay is a cost-effective niche transcriptomics expression profiling method that is not widely used compared to bulk RNA-seq (Subramanian et al., 2017). The most recent L1000 data release in 2021 includes the profiling of over 33,000 small molecule perturbations and over 9,000 genetic perturbations applied to a set of 10 tissue representative core cell lines. Previous L1000 benchmarking studies have focused mostly on evaluating existing L1000 signatures, such as classifying L1000 signatures according to tissue site, cell line subtype, and drug mechanisms of action using deep learning (McDermott et al., 2020); or recovering compound mechanisms of action (MOA) via causal reasoning (Hosseini-Gerami et al., 2023). Additionally, these studies do not include any examination of the CRISPR knockout (KO) perturbations from the latest LINCS L1000 data release. In the past, we have broadly benchmarked various L1000 signature computation methods using transcription factor-target relationships (Evangelista et al., 2022), drug-target associations, and compound structure similarity (Duan et al., 2016). Combining data collected by L1000, RNA-seq, and ChIP-seq studying the same perturbational effect can help evaluate the quality of data processing pipelines for each of these data types because we expect partial agreement between them. Hence, the better agreement we achieve, the better the processing workflow. The breadth of the L1000 data also presents an opportunity to study context-independent effects of different perturbations through the computation of “consensus” signatures for each drug and genetic knockout, which are generally difficult to compute for RNA-seq studies due to the necessarily narrow biological context of most studies.

The Dexamethasone Benchmark (Dex-Benchmark) resource aims to address gaps in the current repertoire of benchmarking tools by providing the community with datasets and template benchmarking workflows that can be adapted to evaluate various transcriptomics datasets, algorithms, workflows, and tools (Fig. 1). The resource is intended to be a “silver standard” benchmark and focuses on datasets related to the drug dexamethasone and its known targets, including the glucocorticoid receptor (NR3C1/GR) (Cole, 2006; Gupta, Awasthi & Wagner, 2007; Wang et al., 2007; Xiang et al., 2021) and the nuclear receptors NR0B1 (DAX1) (Gummow et al., 2006; Yu & Li, 2006) and NR1I2 (PXR) (Lehmann et al., 1998; Luo et al., 2002). By taking advantage of a well-documented drug-target relationship, the Dex Benchmarking Resource can be applied to benchmark methods using real data and evaluate the results against a wealth of previously established results with relative certainty. Moreover, Dex-Benchmark can demonstrate and further elucidate the biological mechanisms underlying dexamethasone treatment and the activation or inhibition of its downstream targets. The Dex-Benchmark resource also uniquely provides benchmarking results for Python-based transcriptomics analysis methods in the form of Jupyter Notebooks, which are available for viewing and download directly from the project’s hosting website. Furthermore, the site uniquely provides benchmarking of various transcriptomics analysis methods for the LINCS L1000 data. Given the scope of this large dataset, it is crucial to identify optimal ways of computing signatures from the L1000 data to take full advantage of the information it contains. The Dex-Benchmark thus uses the known relationship between dexamethasone and its targets as a benchmark for various RNA-seq analysis methods on the L1000 data, including both the computation of individual cell-line specific gene expression signatures and the generation of consensus signatures for identifying cell-line-agnostic effects of drugs. From these results, the benchmarked best practice method is selected to predict drugs that target understudied targets from Illuminating the Druggable Genome (IDG) program (Kropiwnicki et al., 2022).

Figure 1 Overview of the Dex-Benchmark resource.

The website provides compiled data on dexamethasone and known targets NR3C1, NR0B1, and NR1I2, as well as benchmarking code notebooks that can be used to analyze and generate visualizations from the compiled data.

Materials and Methods

Selection of methods and algorithms to benchmark

Differential gene expression methods were selected based on overall usage by the community based on citations, applicability to the data organized and processed, and availability of their implementation as Python libraries. DESeq2 (Love, Huber & Anders, 2014) and limma (Ritchie et al., 2015) are two of the most highly cited packages for differential gene expression analysis of RNA-seq (McDermaid et al., 2019), while the Characteristic Direction (Clark et al., 2014) and MODZ (Subramanian et al., 2017) methods have previously been demonstrated to generate good results for processing the L1000 data. All four of these methods can be implemented in Python or called from existing Python packages.

For enrichment analysis, we chose enrichment analysis tools based on availability, familiarity, and broad usage. Enrichr (Chen et al., 2013; Kuleshov et al., 2016; Xie et al., 2021) and the Python implementation of Gene Set Enrichment Analysis (GSEA) (Fang, Liu & Peltz, 2023; Subramanian et al., 2005) were selected because they are two of the most cited enrichment analysis tools and can be seamlessly integrated into Jupyter Notebooks. Gene expression analysis workflows were evaluated with the ChEA 2022 (Keenan et al., 2019; Lachmann et al., 2010) gene set library from Enrichr. This library contains gene sets representing transcription factor-target associations from ChIP-seq and ChIP-chip experiments manually processed from publications and entries of ChIP-seq studies in NCBI’s GEO. Each gene set corresponds to a specific transcription factor under a specific study condition; associated target genes were determined by mapping the peaks from the original BED files, or by extracting gene IDs from supporting tables of the publications.

Data extraction and compilation

Dexamethasone drug targets were identified from DrugBank (Wishart et al., 2006) and filtered to only those included in the ChEA 2022 (Keenan et al., 2019; Lachmann et al., 2010) gene set from Enrichr (Chen et al., 2013; Kuleshov et al., 2016; Xie et al., 2021), resulting in a final set of three targets: NR3C1, NR0B1, and NR1I2. L1000 Level 5 MODZ signatures for dexamethasone were queried and downloaded from the CLUE.io (https://clue.io) command app (Subramanian et al., 2017). Level 3 beta version expression profiles were downloaded from the LINCS data releases app on CLUE.io. All Level 3 data was filtered to only include dexamethasone perturbations and shRNA knockdown, overexpression, and CRISPR knockout (KO) of NR3C1, NR0B1, and NR1I2 where such data was available. Publicly available RNA-seq datasets related to dexamethasone and the three targets were first compiled through manual queries of GEO (Barrett et al., 2013). GEO series from any species with at least one treatment sample and one control sample were identified and relevant sample IDs were recorded, along with metadata on the original study. Treatment samples were defined as either sample treated with only dexamethasone or samples with one of the relevant targets knocked down or overexpressed.

Benchmarking differential gene expression methods for RNA-seq data

The example notebook for benchmarking differential gene expression methods was applied to RNA-seq count data from four untreated and four dexamethasone-treated samples from peripheral blood mononuclear cells (PBMCs) from GEO series GSE159094 (Northcott et al., 2021), although the notebook works with any RNA-seq count data. Genes were filtered using the filter_by_expr function from edgeR version 3.34.1 (Robinson, McCarthy & Smyth, 2009). A total of five differential gene expression (DGE) methods were benchmarked: PyDESeq2 version 0.3.3 (Love, Huber & Anders, 2014; Muzellec et al., 2022); the characteristic direction (CD) (Clark et al., 2014); limma version 3.48.3 with voom (Law et al., 2014; Ritchie et al., 2015); log2 fold change (FC); and the independent t-test function ttest_ind from the scipy Python package version 1.10.0 (Virtanen et al., 2020). The ranksums function from scipy is implemented but not shown in the RNA-seq benchmarking example due to a small sample size. From each signature, the top 100 up-regulated genes, top 100 down-regulated genes, and the combined 200 top differentially expressed genes as determined by the respective differential expression statistic from each method were submitted to the Enrichr API (Chen et al., 2013; Kuleshov et al., 2016; Xie et al., 2021) for enrichment analysis against the ChEA 2022 gene set library. A hard cut-off was chosen to accommodate methods that do not compute a significance score as well as to evaluate how strongly each method measures the perturbational effects. The enrichment rankings of terms associated with each of the three dexamethasone targets (NR3C1, NR1I2, NR0B1) were recorded for each gene set from each signature computation method and displayed in boxplot format (Table S1). Only terms pertaining to human and mouse data were retained, and one term representing NR3C1 ChIP-seq data from mouse lung endothelial cells was removed due to consistently returning outlier results. For each signature, an averaged running sum bridge plot was also generated to measure the deviation of the cumulative distribution function (CDF) of a given set of gene ranks from the uniform CDF. Briefly, at each rank in a signature, the running sum is incremented if the gene is a member of the comparison target gene set. Genes were ranked from most to least significant based on p-value for the pyDESeq2, limma-voom, and t-test signatures, and by the highest to lowest absolute value of the expression coefficient for the log2 FC and CD methods. The running sum was averaged across all relevant gene sets for each target for visualization.

Benchmarking differential gene expression methods for L1000 data

Two example notebooks were created for benchmarking differential gene expression methods for the L1000 data. The first notebook demonstrates benchmarking of dexamethasone chemical perturbation signatures available from CLUE.io (https://clue.io), and the second notebook demonstrates benchmarking the CRISPR KO signatures for a single target. For each batch representing a unique cell line and a timepoint, treatment samples were the perturbation samples as identified in the Level 3 metadata from CLUE.io, while control samples were all other samples within the same batch. Signatures were computed from the Level 3 L1000 profiles according to the methods described above for the RNA-seq data analysis, except no pyDESeq2 signatures were computed for the L1000 data due to the incompatibility of the signed Level 3 L1000 expression values with the algorithm, and MODZ signatures downloaded from CLUE.io were added to the benchmarking analysis. Due to the larger sample size that is available for the L1000 data, the Wilcoxon rank-sum test was also applied to compute gene-wise changes between the control and treatment samples. Additionally, enrichment rankings for the boxplots and running sums for the bridge plots were first averaged across all batch signatures before they were averaged across all relevant terms.

Computing L1000 consensus signatures

Consensus signatures were computed for 33,609 unique drugs and 7,489 unique CRISPR KO gene targets from the L1000 dataset using three methods: (1) taking the mean expression of each gene across all signatures; (2) taking the median expression of each gene; and (3) taking a weighted average expression of the signatures for each perturbagen where each signature is weighted by a normalized Pearson correlation coefficient computed relative to the other signatures, a method similar to one previously used for computing consensus gene signatures (Smith et al., 2017). Consensus signatures in the example benchmarking notebook and figures were computed from the Level 5 L1000 characteristic direction signatures downloaded from SigCom LINCS (Evangelista et al., 2022).

Benchmarking L1000 consensus signatures

The Pearson correlation coefficient (PCC) for each possible pair of a chemical consensus signature and a CRISPR KO consensus signature was generated, resulting in a similarity matrix of size 33,609 × 7,489. The matrix was filtered to include only drugs with known drug-target relationships documented in Pharos (Nguyen et al., 2017; Sheils et al., 2021), and further filtered to only inhibitor drug-target relationships. Targets were then ranked for each of the resulting 932 drugs from high to low PCC, and an averaged running sum plot was generated for the ranked drug-gene pairs for each drug in a similar manner as described above for the L1000 and RNA-seq data benchmarking.

Results

Differential gene expression analysis benchmarking results for RNA-seq data

The Dex-Benchmark resource provides several Jupyter Notebooks for benchmarking various steps of data processing and analysis of omics datasets. These Jupyter Notebooks are provided as example templates for the community to evaluate the quality of new tools, algorithms, and methods. One of these Jupyter Notebooks was developed to evaluate different methods that compute differentially expressed genes from RNA-seq data. In addition to the known relationships between dexamethasone and its targets, the RNA-seq data analysis serves as a “ground truth” comparison with L1000 transcriptomics. Having matching conditions between RNA-seq and L1000 data is useful for evaluating how the processing of the L1000 compares with mainstream transcriptomics assays. The example notebook compares four dexamethasone treated PBMC samples to four untreated samples from the GEO series GSE159094 (Northcott et al., 2021). Signatures were computed using limma with voom (Law et al., 2014; Ritchie et al., 2015); pyDESeq2 (Muzellec et al., 2022), a Python implementation of DESeq2 (Love, Huber & Anders, 2014); the Characteristic Direction method (Clark et al., 2014); log2 fold change; and the Welch’s t-test. Gene sets from each signature were submitted to Enrichr (Chen et al., 2013; Kuleshov et al., 2016; Xie et al., 2021) for enrichment analysis against gene sets associated with the glucocorticoid receptor (GR/NR3C1) from the ChEA 2022 library (Keenan et al., 2019; Lachmann et al., 2010). In general, the up-regulated gene sets across all signatures rank NR3C1 gene sets higher than down-regulated gene sets and combined up- and down-regulated gene sets (Fig. 2A). This is likely because dexamethasone is a known agonist of GR/NR3C1, and GR/NR3C1 is an activator transcription factor. In terms of signature computation method, for the given example dataset, the limma and pyDESeq2 signatures show comparable top performance compared with the other methods (Fig. 2B), even when broken down by direction (Figs. 2C and 2D).

Figure 2 Example boxplots and bridge plots showing benchmarking results for different signatures generated from dexamethasone perturbation RNA-seq data.

(A) Enrichment rankings of NR3C1 term gene sets from ChEA 2022 for the top up-regulated, down-regulated, or combined up- and down-regulated gene sets across dexamethasone perturbational signatures computed from GSE159094. Black “random” boxes represent randomly sampled gene sets. Boxes are sorted left to right, lowest to highest mean rank. (B) NR3C1 term rankings for dexamethasone perturbational signature gene sets generated by different methods from GSE159094, sorted by mean rank. (C) Boxplot of NR3C1 term rankings for up, down, and combined up/down gene sets generated from each method. Boxes are grouped by method and colored by direction. (D) Boxplot of NR3C1 term rankings for gene sets generated from each method. Boxes are grouped by direction of gene expression and colored by method. (E) A Brownian bridge plot showing target gene set retrieval for RNA-seq dexamethasone perturbation signatures. At each normalized gene rank x for a signature computed using a given method, the running sum y is incremented if the gene is a target in an NR3C1 term gene set from ChEA 2022. Each colored line represents a different method, while the gray line shows the averaged running sum for randomly ordered gene signatures. (F) Leading edge of the Brownian bridge plot from (E).

We then plotted the average deviation of the observed cumulative distribution function (CDF) from uniform according to a method previously used for benchmarking the recovery of drug targets from drug perturbations followed by gene expression (Clark et al., 2014). We evaluated each method based on the leading edge of their respective average bridge plot: a sharper increase, or deviation in the early ranks, is due to more top hits, which indicates the method is capturing a stronger signal. Once again, we see that limma and pyDESeq2 have the sharpest initial peak, indicating that they perform the best at recovering known targets for NR3C1 from ChIP-seq data. Notably, all the methods appear to return non-random results and thus can identify the dexamethasone perturbational signature (Figs. 2E and 2F).

Differential gene expression analysis benchmarking results for L1000 data

The Dex-Benchmark resource uniquely provides code for benchmarking the L1000 chemical perturbation and L1000 CRISPR KO data. The L1000 dataset is a cost-efficient transcriptomics assay that measures the expression of 978 landmark genes and infers the expression of an additional 11,350 genes (Subramanian et al., 2017). The L1000 dataset currently contains over one million unique gene expression signatures, making it a rich source for studying chemical and genetic perturbations across human cell lines. To benchmark the steps in the processing of the L1000 data, signatures were computed from the normalized Level 3 dexamethasone perturbation profiles across 10 different cell lines and treatment timepoints using limma with and without voom (Law et al., 2014; Ritchie et al., 2015), Characteristic Direction (Clark et al., 2014), log2 fold change, Welch’s t-test, and the Wilcoxon rank-sum test. pyDESeq2 was not applied to the L1000 data due to the signed nature of the normalized expression values. Level 5 L1000 MODZ dexamethasone signatures corresponding to the same batches were also downloaded and included for comparison.

Like the RNA-seq data analysis workflow, gene sets from each signature were submitted for enrichment analysis with the ChEA 2022 Enrichr library and the ranks of all NR3C1-associated gene sets were evaluated. In the example notebook, we observe a similar directional signal as before, where up-regulated gene sets across all the signatures are more enriched for NR3C1-associated genes than the down-regulated genes or the combined up/down gene sets (Fig. 3A). This implies that the generated results are indeed reflecting known biology of the dexamethasone-GR relationship. The CD method appears to perform the best for prioritizing target genes in the L1000 signatures, followed by the limma-voom, and log2 fold change methods (Figs. 3B–3D). Except for the t-test, all other methods appear to follow the directional trend where up-regulated genes rank the NR3C1 gene sets the highest (Fig. 3C), consistent with the findings from the RNA-seq data. The genes in each signature were ranked by significance and the leading edge of the bridge plots show the average deviation from the uniform of the CDF for each method. Consistent with the boxplots, the Characteristic Direction method has the most hits for NR3C1-associated genes in the top 2% of the most significantly differentially expressed genes, while the t-test and rank-sum test appear to have the worst performance (Figs. 3E–3F). Notably, the directional gene set pattern seen across the L1000 results, as well as the comparable average rankings with the RNA-seq gene sets, suggests that the L1000 dexamethasone perturbation data produces expected results that are comparable to RNA-seq.

Figure 3 Example boxplots and bridge plots showing benchmarking results for different signatures generated from L1000 dexamethasone perturbation in the A375 cell line.

(A) Enrichment rankings of NR3C1 term gene sets from ChEA 2022 for the top up-regulated, down-regulated, or combined up- and down-regulated gene sets across dexamethasone perturbational signatures computed from the L1000 data. Black “random” boxes represent randomly sampled gene sets. Boxes are sorted left to right, lowest to highest mean rank. (B) NR3C1 term rankings for dexamethasone perturbational signature gene sets generated by different methods from the L1000 data, sorted by mean rank. (C) Boxplot of NR3C1 term rankings for up, down, and combined up/down gene sets generated from each method. Boxes are grouped by method and colored by direction. (D) Boxplot of NR3C1 term rankings for gene sets generated from each method. Boxes are grouped by direction of gene expression and colored by method. (E) A Brownian bridge plot showing target gene set retrieval for L1000 dexamethasone perturbation signatures. At each normalized gene rank x for a signature computed using a given method, the running sum y is incremented if the gene is a target in an NR3C1 term gene set from ChEA 2022. Each colored line represents a different method, while the gray line shows the averaged running sum for randomly ordered gene signatures. (F) Leading edge of the Brownian bridge plot from (E).

Elucidating interactions between dexamethasone targets

One additional interesting result from the benchmarking effort was the verification of the different transcriptional roles of the three known dexamethasone targets. Since NR0B1 and NR1I2 have corresponding L1000 CRISPR KO signatures, we followed the same approach described above for the L1000 dexamethasone signatures to benchmark the NR0B1 and NR1I2 CRISPR KO signatures, comparing them against the associated gene sets for each gene respectively from the ChEA 2022 library. Boxplots displaying the rankings of the associated gene terms show a slight directional bias for both sets of KO signatures: while the down-regulated gene sets from the NR1I2 KO signatures tend to be more enriched for the NR1I2 associated gene sets (Figs. 4A and 4B), and the up-regulated genes from the NR0B1 KO signatures tend to be more enriched for the NR0B1 associated gene sets (Figs. 4C and 4D).

Figure 4 Example box plots showing the rankings of KO gene sets for corresponding LINCS L1000 CRISPR KO perturbational signatures.

(A) Rankings of different NR1I2 KO gene set terms compared to input gene sets obtained from NR1I2 CRISPR KO signatures, grouped by method and colored by the direction of the gene set. A directional effect can be seen, where the down-regulated gene sets from each signature tend to be more enriched for NR1I2 target genes than the up-regulated gene sets. (B) Rankings of NR1I2 KO gene set terms for gene sets generated via all methods for each cell line, grouped by cell line and colored by the direction of the gene set. (C) Rankings of different NR0B1 KO gene sets compared to input gene sets obtained from NR0B1 CRISPR KO signatures, grouped by method and colored by the direction of the gene set. An opposite directional effect to the NR1I2 KO cells is seen here, where the up-regulated gene sets tend to be more enriched for NR0B1 target genes than the down-regulated genes. (D) Rankings of NR0B1 KO gene set terms for gene sets generated via all methods for each cell line, grouped by cell line and colored by the direction of the gene set.

We also examined the overlap between gene sets associated with all three dexamethasone targets and the up- and down-regulated genes from the L1000 weighted average consensus CD signatures for dexamethasone perturbation as well as for the CRISPR KOs of NR0B1 and NR1I2, as the CD method and weighted average consensus method were shown to perform the best at prioritizing target genes from L1000 data. Overlap between the target gene sets and the L1000 up and down genes are visualized with Venn diagrams (Figs. 5A–5I) and SuperVenn diagrams (Fig. S1). In total, 35 genes are overlapping targets of NR3C1, NR0B1, and NR1I2 based on ChIP-seq data from ChEA 2022 (Keenan et al., 2019; Lachmann et al., 2010) (Fig. 6). Of those 35 genes, we found that the genes IL6ST and NRP1 are associated with all three dexamethasone targets and are downregulated under dexamethasone perturbation. Since dexamethasone is a known strong anti-inflammatory drug, its effect on the expression of these genes is likely critical for mediating this clinical effect. NRP1 knockout has been demonstrated to inhibit inflammatory response by mediating IFNγ cytokine activity (Wang et al., 2016), and the IL-6 family of cytokine signaling pathway is known to be majorly involved in inducing inflammation (Tanaka, Narazaki & Kishimoto, 2014; West, 2019). IL6 upregulation has also been correlated with upregulation of NRP1 in pancreatic cancer cells suggesting their mutual involvement with pro-inflammatory pathways and cancer development (Feurino et al., 2007).

Figure 5 Overlapping target genes between consensus L1000 perturbational signature gene sets and NR3C1, NR0B1, NR1I2 target gene sets from ChEA 2022.

(A) Overlap between NR3C1 target genes and dexamethasone consensus up- and down-regulated genes, respectively. (B) Overlap between NR0B1 target genes and dexamethasone consensus up/down genes. (C) Overlap between NR1I2 target genes and dexamethasone consensus up/down genes. (D) Overlap between NR3C1 target genes and NR0B1 KO consensus up/down genes. (E) Overlap between NR0B1 target genes and NR0B1 KO consensus up/down genes. (F) Overlap between NR1I2 target genes and NR0B1 KO consensus up/down genes. (G) Overlap between NR3C1 target genes and NR1I2 KO consensus up/down genes. (H) Overlap between NR0B1 target genes and NR1I2 KO consensus up/down genes. (I) Overlap between NR1I2 target genes and NR1I2 KO consensus up/down genes.

Figure 6 Overlap of target genes of NR3C1, NR0B1, and NR1I2 from ChEA 2022.

Cell types and tissues from which ChIP-seq target genes were identified for each of NR3C1, NR0B1, and NR1I2 are labeled for each circle in the Venn diagram.

Additionally, we found that the gene NFKBIA also belongs to the target gene sets of all three dexamethasone targets and is downregulated under NR0B1 KO. While NR0B1 has long been shown to play a role in adrenal development, and its loss of function is associated with adrenal hypoplasia (Reutens et al., 1999; Suntharalingham et al., 2015), recent research on hepatocytes has suggested that NR0B1 KO leads to inflammatory injury (Yun et al., 2022). While NF-kB activity is a driver of inflammation of immune response processes (Zhang et al., 2021), NFKBIA encodes IkBα, an inhibitor of pro-inflammatory NF-kB activity (Hayden & Ghosh, 2004). Our data support the finding that NR0B1 KO is potentially associated with inflammation via modulation of NFKBIA expression.

On the other hand, FGF1, which is a known anti-inflammatory agent (Fan et al., 2019; Liang et al., 2018), was found to be targeted by all of NR3C1, NR0B1, and NR1I2, and was up regulated in the NR1I2 KO signature. NR1I2 has been implicated in inflammatory diseases (Sun et al., 2022), and our results suggest that one mechanism by which it promotes inflammation may be through downregulating the activity of FGF1. Overall, besides offering a general-purpose benchmark, integrating dexamethasone and its glucocorticoid receptor targets data from RNA-seq, ChIP-seq and L1000 sources helped us zone-in onto the intersection of genes that may be the drivers of the anti-inflammatory effect of dexamethasone.

L1000 consensus signature computation methods benchmarking

Generating consensus drug and CRISPR KO signatures for each perturbation in the L1000 data holds the potential of identifying transcriptomics patterns on a broader, cell-line-independent scale. As such, the Dex-Benchmark site also provides a notebook template for benchmarking L1000 consensus signatures generation with various methods. As a starting point, the resource contains code for generating consensus signatures using the mean, median, and weighted average of the expression values for each gene across all CD signatures corresponding to a single drug or a CRISPR KO gene. The drug signatures and KO signatures can then be compared for their similarity with the Pearson correlation coefficient. This comparison enables the generation of a ranking of the most similar and most opposite gene KO signatures for each drug. These associations can then be evaluated against known drug-target pairs. Known drug-target associations reported in the literature are available from several databases (Ochoa et al., 2022; Wishart et al., 2006; Zhou et al., 2021). For the purposes of this paper, we selected Pharos (Nguyen et al., 2017; Sheils et al., 2021), a centralized data repository and interface for the Illuminating the Druggable Genome (IDG) program (Oprea et al., 2018). We filtered the drug-KO gene pairs to only consider drugs that inhibit a certain target, as we expect that the perturbational signature of an inhibitory drug to be more like the knockout signature of its target gene. When the drug-KO gene pairs are ranked from high to low PCC for consensus signatures computed using each of the three benchmarked methods, all three methods perform better than random, indicating the consensus signatures are all capable of extracting drug-target knowledge from the L1000 data even across multiple cell lines. The mean and weighted average methods perform slightly better than the median in prioritizing known drug-target pairs (Fig. 7). While there are many other ways to establish consensus signatures for the L1000 data, the benchmark provides an objective path to assess the quality of such consensus signatures.

Figure 7 Example Brownian bridge plots showing the running sum for L1000 inhibitory drug-gene knockout consensus signature pairs.

(A) Each colored line shows the averaged running sum for consensus CRISPR KO gene signatures that are most like a given drug perturbation signature, while the gray line shows the averaged running sum for randomly ordered CRISPR knockout genes. At each normalized gene rank x, the running sum y is incremented if the gene knocked out in the signature is a known inhibitory target of the drug. All three non-random methods perform comparably, although the median slightly underperforms both weighted and unweighted averages. (B) The leading edge (x <= 0.01) of the plots shown in (A). The median consensus signature method is more clearly seen to be capturing less target genes on average in the top 1% of the most similar gene signatures for each drug compared to the mean and weighted average methods.

Predicting drugs for understudied targets

Since the L1000 consensus signature methods were shown to be able to prioritize known drug-target relationships from Pharos just based on the similarity between a drug signature and a target gene KO signature, there is potential to predict yet-unknown drug-target relationships using the L1000 data in conjunction with IDG data. For each of the 161 targets from the 2022 version of the IDG understudied proteins list which overlap with the L1000 KO genes, we computed the PCC between the weighted average consensus of the KO signature for the target and all the chemical perturbation weighted average signatures. The top 100 most similar and most opposite drug signatures for each target are computed (Tables S2 and S3). These compounds are prioritized as potential direct or indirect inhibitors or activators of the under-studied targets. Clearly, further experimental, and orthogonal computational validation is needed to truly confirm these associations.

The Dex-Benchmark website

The Dex-Benchmark resource is available as a web interface hosted on GitHub pages at https://maayanlab.github.io/dex-benchmark (Fig. 8A). Example code notebooks may be viewed and downloaded from the “Code” page (Fig. 8B), while spreadsheets of all relevant datasets can be found on the “Data” page (Fig. 8C). Examples and explanations of boxplot and running sum plots from the benchmarking notebooks, including those described here, can be found under the “Figures” page (Fig. 8D). Finally, a “Stats” page provides statistics for the resource such as the total number of data sources, notebooks, and computed signatures (Fig. 8E).

Figure 8 Screenshots of the Dex-Benchmark resource website.

(A) Home page of the Dex-Benchmark site. (B) The code page lists all example benchmarking notebooks and HTML files. (C) The datasets page lists all compiled datasets related to dexamethasone and its genetic targets. (D) The figures page displays example figures from the benchmarking notebooks. (E) The statistics page displays statistics for the datasets available in the resource.

Discussion and conclusions

The Dex-Benchmark resource enables the benchmarking of various methods and algorithms for analyzing transcriptomics data by providing curated datasets and Jupyter Notebooks for performing analyses and visualization with real biological and pharmacological data. While the resource does not cover all existing analysis methods, it provides a framework for the community to add these. The resource is unique compared to other benchmarking efforts because it brings together Python-based tools, and it provides a “silver standard” relationship between several independent omics data types with some expected concordance.

By making all the data and code openly available on GitHub, we hope that the Dex-Benchmark will facilitate the fair evaluation of new tools by the bioinformatics community. Overall, we found that across different cell lines, the L1000 dexamethasone perturbation data produced similar differentially expressed gene signatures to RNA-seq datasets for dexamethasone perturbation. Perhaps more importantly, we found that both methods were able to reveal the biologically expected results based on known relationships between dexamethasone and its known targets NR3C1, NR0B1, and NR1I2. However, the benchmarking results suggest that different analysis methods should be used for processing data produced by the RNA-seq and L1000 assays. Consistent with the most applied methods, our preliminary results suggest that limma (Ritchie et al., 2015) and pyDESeq2 (Love, Huber & Anders, 2014; Muzellec et al., 2022) perform well in the analysis of RNA-seq data. On the other hand, we found that the Characteristic Direction (CD) method performs the best at identifying differentially expressed genes in the L1000 data. This result is consistent with previous work showing that the CD method can effectively extract drug class information from the L1000 data (Duan et al., 2016; Niepel et al., 2017; Wang et al., 2018). One potential explanation for the different performance of the methods between the RNA-seq data and the L1000 data is that the L1000 expression values are already normalized at the feature level, and the expression levels can be both negative and positive. Further normalization during DGE analysis may obscure any existing signals in the data, making the gene rankings appear to be more random for some methods. Lonnstedt & Nelander (2017) have proposed methods for normalizing the Level 4 L1000 data, but further research related to the performance and biases of Level 3 L1000 data may be warranted. We also demonstrate how to benchmark consensus signatures for the L1000 perturbations and show that drug-target information extracted from the consensus signatures aligns with current knowledge from Pharos (Nguyen et al., 2017; Sheils et al., 2021). As such, we believe that further benchmarking of L1000 data analysis methods can lead to even greater understanding of understudied targets that have already been profiled via CRISPR knockouts followed by the L1000 assay.

Our analyses of the different dexamethasone targets also led to a deeper understanding of the transcriptional role of each of these targets. NR1I2 is well-documented as a transcriptional activator (Ihunnah, Jiang & Xie, 2011), while NR0B1 has been shown to have a more negative transcriptional role in both humans and mice (Susaki et al., 2012; Fujii et al., 2015). Our results suggest that NR1I2-associated target genes based on ChIP-seq tend to be down-regulated when NR1I2 is knocked out, while NR0B1-associated genes tend to be up-regulated when NR0B1 is knocked out, which is consistent with their functions described in the literature. Additionally, in comparing the overlap between target genes for each of NR3C1, NR0B1, and NR1I2 with the up and down regulated genes from the L1000 consensus signatures, we identified further clues into how dexamethasone and its targets interact to mediate inflammatory responses. Specifically, we found support for the anti-inflammatory effects of dexamethasone perturbation via the downregulation of IL6ST and NRP1, which are targets of all NR3C1, NR0B1, and NR1I2. Our analysis suggests that NR0B1 loss of function may promote inflammation via dysregulation of NFKBIA and consequently the entire NF-kB pathway, while NR1I2 knockout may have anti-inflammatory effects through the up-regulation of FGF1. To the best of our knowledge, these associations have yet to be explored in the literature and may warrant further experimental investigation. Overall, our results from a preliminary benchmarking of L1000 consensus signatures are promising in demonstrating the utility of the L1000 CRISPR KO data, which was made available in the most recent 2021 LINCS data release, for further elucidating the biological functions of known and novel targets.

Altogether, by combining raw and processed data from transcriptomics of drug- and TF-knockout induced expression changes, with data from ChIP-seq studies that profiled the same transcription factors, we can establish unbiased silver standard benchmarks that can be used to evaluate the quality of many central bioinformatics tools. At the same time, such data integration efforts can discover novel insights about detailed molecular mechanisms that govern key intracellular biological and pharmacological processes.

Supplemental Information

Supplemental Information 1 Supervenn diagrams showing overlap between consensus signature gene sets and all three target gene sets.

(A) Overlap between dexamethasone up/down gene sets and NR3C1, NR0B1, NR1I2 target gene sets. Each row indicates a different set. The column labels at the top display the number of overlapping sets, while the column labels at the bottom display the number of overlapping items. (B) Supervenn diagram showing overlap between NR0B1 CRISPR KO consensus up/down genes and all three target gene sets. Column labels are the same as in (A). (C) Supervenn diagram showing overlap between NR1I2 CRISPR KO consensus up/down genes and all three target gene sets. Column labels are the same as in (A) and (B).

Click here for additional data file.

Supplemental Information 2 ChIP-seq target gene set terms for dexamethasone targets from ChEA 2022 library.

Click here for additional data file.

Supplemental Information 3 Top 100 predicted drugs with L1000 consensus signatures most opposite to each IDG target, based on Pearson correlation coefficient.

Click here for additional data file.

Supplemental Information 4 Top 100 predicted drugs with L1000 consensus signatures most similar to each IDG target, based on Pearson correlation coefficient.

Click here for additional data file.

Additional Information and Declarations

Competing Interests

Author Contributions

Data Availability

The authors declare that they have no competing interests.

Zhuorui Xie performed the experiments, analyzed the data, prepared figures and/or tables, authored or reviewed drafts of the article, and approved the final draft.

Clara Chen performed the experiments, analyzed the data, authored or reviewed drafts of the article, and approved the final draft.

Avi Ma’ayan conceived and designed the experiments, analyzed the data, authored or reviewed drafts of the article, and approved the final draft.

The following information was supplied regarding data availability:

The code is available at GitHub and Zenodo:

- https://github.com/MaayanLab/dex-benchmark.

- Xie, Zhuorui. (2023). Source code and file for a publication titled: “Dex-Benchmark: datasets and code to evaluate algorithms for transcriptomics data analysis” (10-11-2023). Zenodo. https://doi.org/10.5281/zenodo.8433206.

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
