# Peer review of "Dex-Benchmark: datasets and code to evaluate algorithms for transcriptomics data analysis"

_PeerJ, doi:10.7717/peerj.16351_

## Round 0.1 · original submission · Major Revisions

Thank you for your patience during the review process.

Both reviewers were enthusiastic about the work and feel that this resource will be valuable to the community. However, there were concerns raised, the major ones being focused around difficulty in following the overall flow of the work and associated confusion as to the interpretation of the presented results and broader findings. While I did think the writing was clear, there are a significant number of "moving parts" in this work, which I do agree with the reviewers that this complexity makes it quite challenging to fully understand what is being done as well as how to interpret the findings.

While there is no requirement to restructure the manuscript as suggested by reviewer 1, the above mentioned challenges in understanding and interpretation suggest that it would be extremely valuable to adjust or reorganize the manuscript so as to make it easier for the reader to better follow and appreciate the work performed. This is the driver behind the "major revisions" decision.

Reviewer 1 ·

Basic reporting

This submitted manuscript has a very well deserved goal, i.e. to set a standard for analysis of transcriptomic data and codes. As the authors pointed out, there are multiple methods for sequence alignment, data normalization, clustering, DGE determination and gene set enrichment. If there is a platform to select the best methods, it will be widely used. The regulation of genes by dexamethasone was also a very good choice.

Experimental design

The manuscript has enormous amounts of data (both sources as well as generated) and derived datasets in addition to references and projections. It could be easily divided into at least two manuscripts, the first one dealing with the original goal, i.e. to find the best tools to analyze only data generated by single cell RNA sequencing and microarray derived from different sources, not to make it complicated and confusing with the addition of L1000 derived data.

In order to decide which method is the best one, the authors collected big data generated using microarray and single cell RNA, of samples treated with/without dexamethasone. In addition they also collected enormous amount of data from L1000 and combined both. That made it extremely confusing and hard to follow the results and conclusions.

Validity of the findings

A clear description of selection of a tool in each category with explanation of why that was the valid choice biologically, would have been ideal. I did not see any criterion set for that. Enrichr analysis was used with no explanation given why that was the best method for biologically relevance for choosing a method in each category such as (a) sequence alignment, (b) data normalization, (c) clustering, (d) DGE and (e) gene set enrichment.

I would recommend to stick to the goal suggested in the beginning, i.e. to create a platform to decide which method is ideal for a systematic analysis in a specific situation. It is nice to have the pre-analyzed public data from different sources using different methods to use as standards. However, I won’t recommend this manuscript in this form for publication. It is extremely confusing because of juxtaposition of lots of data from microarray, single cell RNA sequening as well as from L1000, without clear explanation of results and conclusions. Combination of large scale transcriptomic data and L1000 generated outputs appears incoherent to me to understand the biological role of dexamethasone.

Additional comments

Let this manuscript present the means and reasons to select an ideal method for each of these: (a) sequence alignment, (b) data normalization, (c) clustering, (d) DGE and (e) gene set enrichment with proper biological explanation. Their next manuscript should deal with the data from L1000 exclusively.

·

Basic reporting

This paper describes an innovative strategy to benchmark differential gene expression pipelines by exploiting the highly characterized downstream gene expression changes in response to dexamethasone treatment. The authors provided sufficient literature background for the current state of differential gene expression techniques and motivated the need for such a tool effectively. The language is clear and concise aside from a few trivial grammatical errors (I would encourage another readthrough by the authors). The plots are fairly clear with one major suggestion for presentation: A colour legend would be much more clear and appropriate for the plots for Fig 2C and 2D instead of using the ":up" vs "combined" etc. method of annotating groups (similar to Fig 2E and 2F). The same applies for figures 3C, 3D, and figures 4A-4D.

Experimental design

The study design employed by the authors is simple and efficient, as is appropriate for the description of a bioinformatic tool. The application of the dexamethasone benchmark to different types of datasets, including for understudied proteins is laudable. A minor comment I have is for the results presented in figure 5 and 6, more explanation is needed regarding the usage of ChIP-Seq data to infer targets of NR3C1. While the citations are helpful, it would be additionally informative to have a couple of sentences in the paper as to how those targets are annotated using ChIP data. It is not immediately clear how the DNA-binding data from ChIP-seq is used since the three targets of dexamethasone are not DNA-binding.

Validity of the findings

The results of the described study are novel and innovative, with sufficient care taken to ensure wide usability through the published jupyter notebooks. All the code and data is made available through github.

---

## Round 0.2 · accepted · Accept

Thank you for addressing the reviewers' concerns. Congratulations again!

Reviewer 1 ·

Basic reporting

The authors have modified the manuscript to my satisfaction.

Experimental design

In the revised manuscript the experimental design is much clear.

Validity of the findings

Validation of the findings has been corroborated with literature citation.

·

Basic reporting

The authors have successfully revised my proposed changes in the paper.

Experimental design

No comment.

Validity of the findings

No comment.